# An Ecological Function Approach to Managing Harmful Cyanobacteria in Three Oregon Lakes: Beyond Water Quality Advisories and Total Maximum Daily Loads (TMDLs)

Eric S. Hall [1,*], Robert K. Hall [2], Joan L. Aron [3], Sherman Swanson [4], Michael J. Philbin [5], Robin J. Schafer [6], Tammy Jones-Lepp [7], Daniel T. Heggem [8], John Lin [8], Eric Wilson [9] and Howard Kahan [2]

1   USEPA Office of Research and Development, NERL, Systems Exposure Division (SED), Ecological and Human Community Analysis Branch, Research Triangle Park, NC 27709, USA
2   USEPA Region IX, WTR2, 75 Hawthorne St., San Francisco, CA 94105, USA; hall.robertk@epa.gov (R.K.H.); kahan.howard@epa.gov (H.K.)
3   Aron Environmental Consulting, 5457 Marsh Hawk Way, Columbia, MD 21045, USA; joanaron@ymail.com
4   Ecology, Evolution and Conservation Biology, University of Nevada, 1664 N. Virginia St., Reno, NV 89557, USA; sswanson@cabnr.unr.edu
5   U.S. Dept. of the Interior Bureau of Land Management, Montana/Dakotas State Office, 5001 Southgate Drive, Billings, MT 59101, USA; mphilbin@blm.gov
6   University of Puerto Rico, Río Piedras Campus, 14 Ave. Universidad, Ste. 1401, San Juan, PR 00925-2534, USA; robin.schafer@upr.edu
7   USEPA Office of Research and Development, NERL, Exposure Methods and Measurement Division (EMMD), Environmental Chemistry Branch, Las Vegas, NV 89119, USA; tjoneslepp@gmail.com
8   USEPA Office of Research and Development, NERL, Systems Exposure Division (SED), Ecosystem Integrity Branch, Las Vegas, NV 89119, USA; motoheggem2@gmail.com (D.T.H.); lin.john@epa.gov (J.L.)
9   Gulf Coast STORET, LLC, 11110 Roundtable Dr., Tomball, TX 77375, USA; ericwilson@gulfcoaststoret.com
*   Correspondence: hall.erics@epa.gov; Tel.: +1-919-541-3147

**Abstract:** The Oregon Department of Environmental Quality (ODEQ) uses Total Maximum Daily Load (TMDL) calculations, and the associated regulatory process, to manage harmful cyanobacterial blooms (CyanoHABs) attributable to non-point source (NPS) pollution. TMDLs are based on response (lagging) indicators (e.g., measurable quantities of NPS (nutrients: nitrogen {N} and phosphorus {P}), and/or sediment), and highlight the negative outcomes (symptoms) of impaired water quality. These response indicators belatedly address water quality issues, if the cause is impaired riparian functions. Riparian functions assist in decreasing the impacts of droughts and floods (through sequestration of nutrients and excess sediment), allow water to remain on the land surface, improve aquatic habitats, improve water quality, and provide a focus for monitoring and adaptive management. To manage water quality, the focus must be on the drivers (leading indicators) of the causative mechanisms, such as loss of ecological functions. Success in NPS pollution control, and maintaining healthy aquatic habitats, often depends on land management/land use approaches, which facilitate the natural recovery of stream and wetland riparian functions. Focusing on the drivers of ecosystem functions (e.g., vegetation, hydrology, soil, and landform), instead of individual mandated response indicators, using the proper functioning condition (PFC) approach, as a best management practice (BMP), in conjunction with other tools and management strategies, can lead to pro-active policies and approaches, which support positive change in an ecosystem or watershed, and in water quality improvement.

**Keywords:** cyanobacteria; ecological function; ecosystems; harmful cyanobacterial bloom (CyanoHAB); proper functioning condition (PFC); total maximum daily load (TMDL); non-point source (NPS); point source (PS); Oregon Department of Environmental Quality (ODEQ); best management practice (BMP)

---

## 1. Introduction

Watersheds are complex ecosystems [1]. Nature is not static, but adjusts and adapts to natural climatic and anthropogenic stresses [2,3]. In aquatic environments, not all water pollution is from an external input. Pollution can come from the materials stored in riparian areas, and wetlands, due to their attributes, physical processes, and functions [4]. When an abundance of nutrients, increased warmth, higher salinity, and light are available, harmful cyanobacterial blooms (CyanoHABs) can occur [5]. Regulating water pollution is a key U.S. Clean Water Act (CWA) tool [6]. Point source (PS) pollution control strategies were very effective during the early decades of the United States Environmental Protection Agency (USEPA), when success was attributed to regulatory control structures capable of "breaking" the pathway of point source (PS) contamination to an endpoint cohort (i.e., humans, fish, etc.), and attaining water quality standards [6].

As the emphasis for pollution control moved from point sources to non-point sources, success has been sporadic [7], because a singular solution, or control structure(s), are not useful approaches for stemming non-point source (NPS) pollution. The Water Resources Development Act of 2007 (Available online: https://www.govinfo.gov/content/pkg/PLAW-110publ114/html/PLAW-110publ114.htm; accessed: 28 May 2019), specifies that federal water resources investments shall, "reflect national priorities, protect the environment, maximize sustainable economic development, avoid excessive use of floodplains and flood-prone areas, and protect and restore the functions of natural systems". For PSs, the CWA provided a framework (Sections 303(d), 305(b), 319) for identifying best management practices (BMPs) needed to reach required water quality standards. Once a waterbody is designated as impaired, per CWA Section 303(d), a total maximum daily load (TMDL) source assessment, load allocation, and in most states an implementation plan document is required. Creating a non-point source TMDL is often challenging, because a plan requires feasible, and accurate, accounting of allowable non-point source pollutant loadings and regulatory control methods. Leading indicators measure the performance of current and future ecological conditions that drive pollution [8,9]. The lack of success in addressing NPS pollution is partly because water quality, and aquatic organism measures, are typically lagging indicators [8–10].

A lagging indicator may eventually respond (e.g., excess sediment, nitrogen (N), and phosphorus (P)), but not soon enough to guide decisions needed to ensure progress [10–12]. Land cover and land use, in non-riparian watershed areas, can drive erosion or release and transport pollutants, into an expanded network of ditches and drains. Watershed vegetation is critical for slowing down hydraulic forces, which are magnified through peak hydrology, and efficient channel forms, and greatly influences the movement of eutrophication nutrients, such as nitrogen and phosphorus. Drivers (leading indicators) of physical function identify early interventions to prevent the loss of assimilation processes, and water quality deterioration. The objective of this study was to show that interdisciplinary, qualitative assessments of riparian and watershed function, and biophysical alterations at a local scale, such as the proper functioning condition (PFC) approach, can assist resource managers in prioritizing adaptive management practices (i.e., objective implementation, monitoring, etc.) [13–15].

### 1.1. Ecosystem Function

Bernhardt et al. [16], noted that most of the billions of dollars [17,18] spent to enhance water quality and improve in-stream habitat lacks the appropriate data and information to evaluate the ecological

effectiveness of restoration activities. Therefore, it is essential to monitor vegetation, hydrology, soils, landform, and water quality parameters, as part of an ecological restoration project, to collect data/information essential for State and Tribal environmental managers to prioritize areas of focus and concern [19]. Ecosystem function is defined as a state of resiliency, which allows a riparian wetland system to remain intact during a 25 to 30-year flood event, and sustains an ecosystem's ability to produce ecosystem values related to physical and biological attributes [20,21]. Water quality and biological communities are affected if a stream's riparian function is impaired. A stream's shape evolves over time, in response to flow and sediment loads [22]. Increased volume and intensity of runoff, or improper watershed or riparian management, often leads to accelerated stream erosion, which may result in streams becoming straighter, steeper, and deeper through incision, or wider and shallower, through bank erosion and aggradation [23]. These reduce pool habitat, and fish cover, and destroy riparian vegetation [14,20].

Proper functioning condition (PFC) refers to how well the physical processes within a stream and wetland riparian area can sustain a state of resiliency [14,15,20,21]. This resiliency allows an area to provide desired and valued ecosystem services (e.g., fish habitat, livestock, and/or wildlife forage, water purification, carbon storage, and nutrient cycling) over time [21]. Functional ecosystems are resilient to disturbances (e.g., floods), in contrast to nonfunctional ecosystems, which fail to buffer against surges in flow from upland and upstream inputs. This is because ecosystems are interconnected communities of vegetation, hydrology, soils, landform, and micro-organisms linked by physical and chemical interactions [14]. For example, a change in vegetation impacts a lentic (still, fresh water) or lotic (rapidly-moving, fresh water) ecosystem's physical functions (assimilative capacity and bank stability). If vegetation deteriorates, and the system is not functioning properly, water quality will degrade. When the ecosystem is functioning, and moving in a positive direction, water quality will improve [9,10,20]. Therefore, it is important for a riparian system to have not only vegetation, but it must have the right kind of obligate stabilizing vegetation, appropriate for the setting [24], to protect banks [15,25,26].

## 1.2. Harmful Cyanobacterial Blooms

While cyanobacteria are one of the most primitive and pervasive forms of life in the aquatic environment, the over stimulation, and proliferation in eutrophying fresh waters, of cyanobacterial species capable of producing powerful toxins, can be detrimental to water bodies. Some of the negative impacts are: economic—from the loss of income due to the loss of recreational use, and loss of fishing resources; human health—from exposure to cyanobacterial toxins via drinking water, and; ecological—from hypoxia and fish kills, cattle and other domestic animal poisonings, from lakes and ponds that are afflicted with CyanoHABs [27]. Health-threatening cyanobacterial toxins are caused by a variety of species of cyanobacteria, with a lot of attention being focused on *Microcystis* species, which release the group of toxins known as microcystins [28]. Besides *Microcystis*, other species of CyanoHABs are rising to predominance in both the US, and globally, for example, the filamentous cyanobacteria *Dolichospermum planctonicum* (*syn: Anabaena plactonica*, [29]) and *Aphanizomenon flos-aquae* [30–32].

## 1.3. Oregon Department of Environmental Quality's Use of TMDLs to Manage CyanoHABs

The northwestern U.S. state of Oregon's Department of Environmental Quality (ODEQ) uses the TMDL process to manage CyanoHABs. A TMDL calculates the maximum amount of a pollutant that a waterbody can receive and still safely meet water quality standards. Harmful cyanobacteria in Oregon lakes are more strongly correlated with variation in total phosphorus, than whether the waterbody has the right amount of nitrogen relative to the amount of phosphorus [33]. TMDLs were designed to reduce sediment, nitrogen, and phosphorus. Because of the special nitrogen fixing properties of some species of cyanobacteria (e.g., *Dolichospermum planctonicum (syn: Anabaena plactonica)* and *Aphanizomenon* [29,32]), the reduction of phosphorus is the most effective means of reducing CyanoHABs [33–35]. TMDL levels

were set individually for each targeted waterbody using information from the Atlas of Oregon Lakes [35], and benchmark water quality parameters established by USEPA. ODEQ combined its TMDL based strategy, with a solid surveillance and monitoring program for CyanoHABs [36], water quality and drinking water quality standards, waste water permitting, and some targeted educational outreach, including health advisories for CyanoHABs (Figure 1).

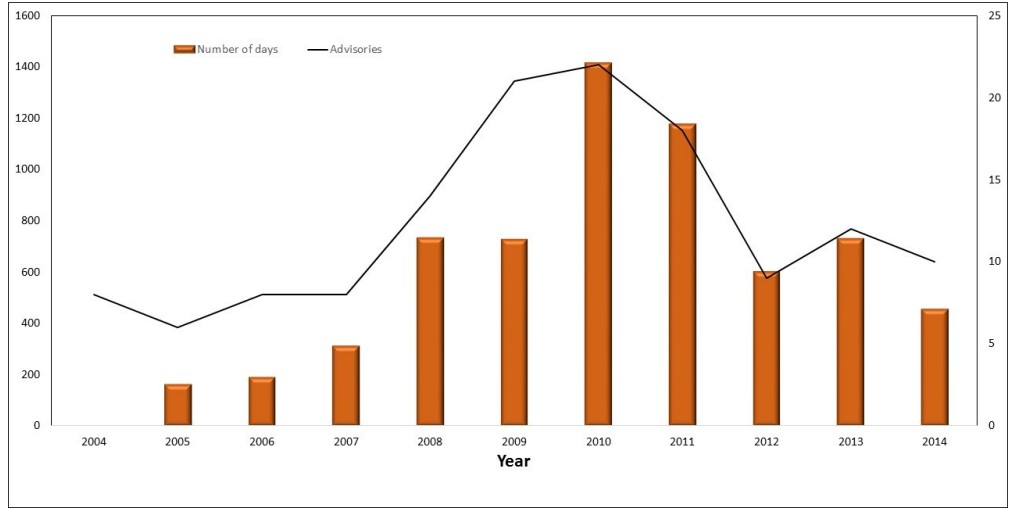

**Figure 1.** Oregon Department of Environmental Quality (ODEQ)'s number of harmful cyanobacterial blooms (CyanoHAB) waterbody advisories (line), and number of cumulative days per year (bars) under an advisory. (Available online: http://www.oregon.gov/oha/ph/HealthyEnvironments/Recreation/ HarmfulAlgaeBlooms/Pages/index.aspx; accessed: 28 May 2019). <u>Note</u>: There were no CyanoHAB advisory days in Oregon for 2004.

As seen in Figure 1, ODEQ has had difficulty in reducing the number of advisories, and the number of waterbody days under an advisory, to below 2005 levels (when warnings were initiated). As a result, ODEQ used its regulatory authority to develop an early warning system to post health advisories, informing the public of the danger of increased exposure to CyanoHABs, and to "break" the exposure pathway (prevent contact with CyanoHABs in water and/or wildlife exposed to CyanoHABs). ODEQ and the USEPA marked the early warning system as a sign of success in solving immediate health problems. However, these existing protocols miss an opportunity to implement a long-term, sustainable solution to reduce the number of advisories. A sustainable solution is outlined for the Oregon Tenmile Lakes TMDL [37], which states that "improving watershed stream and wetland riparian functions was important in curtailing sediment and nutrient loads". This study recommends using the ODEQ TMDL approach, in conjunction with, time-series photography, CyanoHAB advisories, and the PFC methodology, as a preferred (best) practice, instead of using TMDLs without complementary approaches. This study illustrates the use of the proper functioning condition (PFC) methodology in assessing the state of the ecosystem of three Oregon lakes, and in developing a resource management plan that can enhance those ecosystems.

## 2. Methods

Time-series, remote-sensing data from Google Earth was used to evaluate the impact of different land use policies on the region, due to the consistency of coverage within the study area (the Tenmile Lakes (natural lake), Lemolo Lake (impoundment), and Diamond Lake (natural lake)). ODEQ has a well-developed surveillance and monitoring program [36], in coordination with educational outreach, waste water permitting, water quality and drinking water quality standards, and health advisories for CyanoHABs, linked with its TMDL approach. This was augmented with a proper functioning

condition (PFC) stream and wetland riparian area assessment protocol [15,20], conforming to the United States Department of Agriculture's (USDA's) United States Forest Service (USFS) process.

PFC is a qualitative assessment process, which is implemented by a multidisciplinary/interdisciplinary team (ID team) [14,15,20,38]. The PFC assessment framework is a consistent, qualitative, science-based approach for considering stream and wetland hydrologic, vegetative, and geomorphic attributes and processes, at a point in time [15]. PFC is also an appropriate starting point for determining, prioritizing, and collecting information about riparian resources, developing monitoring needs [37], and providing context for quantitative data. An ID team must understand stream dynamics and its potential natural condition (PNC: i.e., the highest attainable ecological status of a riparian area without consideration of economic, political, or social constraints), and use their professional experience and judgment to accurately complete a qualitative assessment [38]. Use of quantitative data and techniques, i.e., field measurements [25,39] or remote sensing [20], is encouraged for individual or team calibration, or where opinions may differ [15].

PFC is used to describe both "an assessment process, and a defined, on-the-ground, condition of a riparian-wetland area". A PFC rating, having 17 attributes for a lotic ecosystem [15], and 20 attributes for a lentic ecosystem [20], relates how well the physical stream/riparian processes are functioning in comparison to its PNC. By focusing on physical functioning (hydrology, vegetation, soil, and landform attributes) the PFC protocol is designed to yield information about the biology of the plants and animals dependent on the riparian-wetland area [14,20].

After a PFC assessment is completed, it provides a rating for the "riparian-wetland area" of either "proper functioning condition", "functional-at-risk", "apparent trend", or "non-functional". Stream and wetland riparian areas in "proper functioning condition" (PFC) status sustain a state of resiliency appropriate for local ecological potential, and provide ecosystem services (e.g., wildlife and fish habitats, diminished flood impacts, and improved water quality). PFC refers to how well stream and wetland riparian physical processes can sustain a state of resiliency after a natural or anthropogenic disturbance [14,15,20,21]. A resilient, properly functioning, ecosystem can assimilate stressors and produce values related to both physical and biological attributes.

The "functional-at-risk" rating refers to a riparian area that is functioning, but with an existing soil, water, or vegetation attribute making it susceptible to degradation. The "apparent trend" rating is an assessment of the direction of change (e.g., upward or downward) in condition, either towards or away from the PNC or the PFC [14,20]. The apparent trend is determined by comparing the present condition with previous photos, trend studies, inventories, other documentation, or personal knowledge. The "non-functional" rating indicates the stream and wetland riparian area are in a degraded state.

*2.1. Study Area*

Streams differ in their potential to produce habitats, biota, and water quality for beneficial uses (i.e., swimmable, fishable, drinkable, etc.). Water quality standards (WQS) determine the allowable concentration of pollutant loads within the waterbody. The question is how to reduce pollutant loads, when many streams are themselves the source of sediment and/or nutrients, and this points to the importance of having a water quality management plan, as was developed for the Tenmile Lakes watershed [40], and Diamond Lake and Lemolo Lake [41], to mitigate those pollutant sources. The three lakes in Oregon that were studied are shown in Figure 2, and their general characteristics are provided in Table 1.

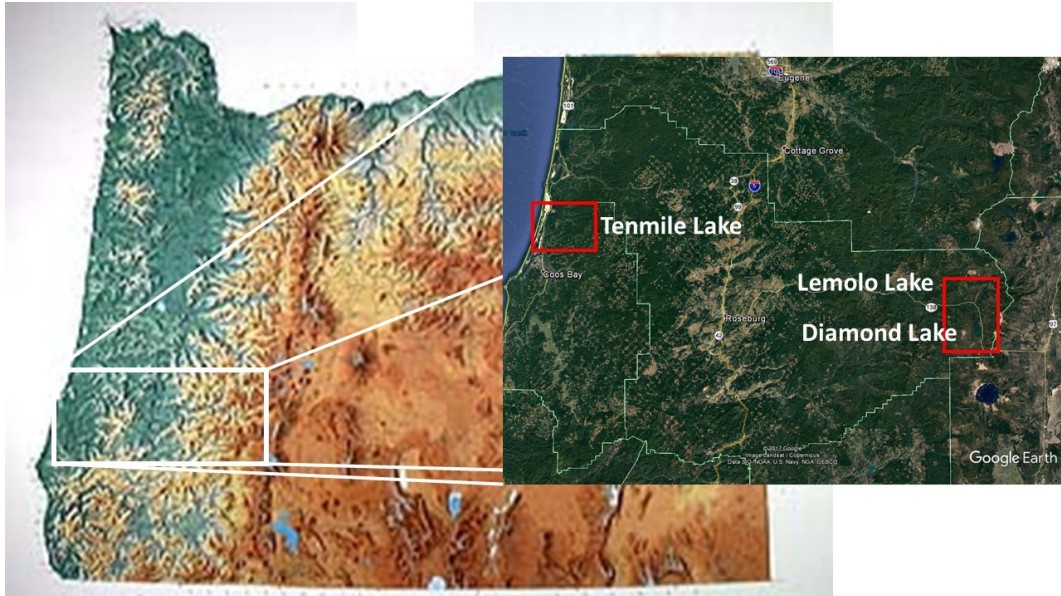

**Figure 2.** Map of the Lemolo Lake, Diamond Lake, and the Tenmile Lakes study areas. The larger image is a Landsat image from 15 July 2014. The inset image is from Google Earth.

**Table 1.** General characteristics of Diamond Lake (North Umpqua Watershed), Lemolo Lake (North Umpqua Watershed), and Tenmile Lakes (Tenmile Creek Watershed) in Oregon. Source: Reference [42].

| Lake | Area (Acres) | Maximum Depth (Feet) | Average Depth (Feet) | Shoreline Length (Miles) |
|---|---|---|---|---|
| Diamond | 3214 | 52 | 24 | 9 |
| Lemolo | 450 | 100 | 30 | 7.9 |
| Tenmile | 1627 | 22 | 10 | 22.9 |

### 2.1.1. Oregon Southwestern Coast and Range: Diamond Lake and Lemolo Lake

The North Umpqua River is a tributary to the Umpqua River, which headwaters are in the low-relief province of the Cascade Mountains of southwestern Oregon (Figure 2). This portion of the Cascades is underlain by highly permeable Pliocene and Quaternary lava flows, which have low rates of surface-water runoff and sediment transport [43]. Forest management prior to 1970 predominantly consisted of extensive "clear cutting" [44]. After 1970, the forest management approach changed to smaller patchwork "clear cuts" (Figure 3). The United States Forest Service (USFS) ecosystem function assessment indicates that stream channel stability in the managed areas of the North Umpqua River watershed did not significantly differ from those in unmanaged drainages [45]. The USFS determined these streams were relatively stable, and in proper functioning condition [45]. Around 2005, the forest management strategy was changed to "forest thinning", as seen around Lemolo Lake (Figure 3A,B).

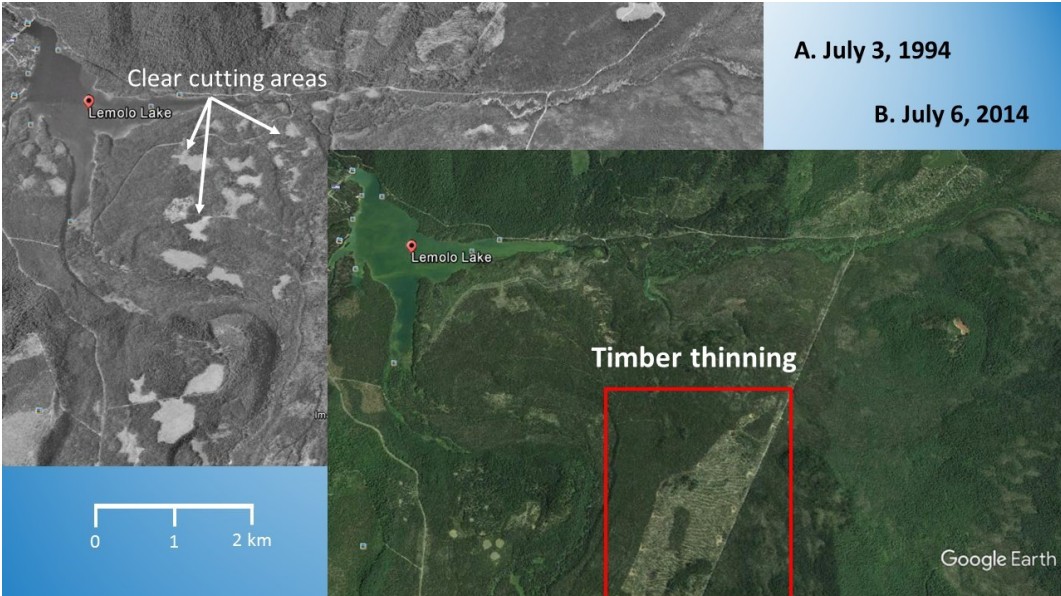

**Figure 3.** United States Department of Agriculture (USDA) black and white, and color aerial images, of the Lemolo Lake area, Oregon. Images were taken: (**A**) 3 July 1994 (black and white), and (**B**) 6 July 2014 (color). Bright patches in image A indicate areas of clear cutting. The red boundary in image B is an area currently being thinned. Source: Images obtained from Google Earth.

Phosphorus is associated with the decomposition of the volcanic rock of the Cascade Mountains, which have been "clear cut" (Figure 3A) [42]. Remote sensing analysis indicated there was a change in forest management from "clear cutting" (Figure 3A), prior to 2005, to "forest thinning" (Figure 3B).

Diamond Lake (Figure 4), a nitrogen-limited eutrophic lake, peaking at 0.5 mg/L total nitrogen (N), and nearly 0.2 mg/L ammonium-N in 2007 [41], is an important source of water and nutrients to the North Umpqua River [46]. Diamond Lake is listed as a "water-quality limited" waterbody (pH, CyanoHABs). Diamond Lake is naturally high in phosphorus, as high as 0.060 mg/L [41]. CyanoHABs have occurred in Diamond Lake since 2001 [46], resulting in closures of recreational water facilities. Increased CyanoHABs have been attributed to increased populations of the invasive, zooplankton-eating, Tui Chub fish [47]. Tui Chub feed on zooplankton that would otherwise control phytoplankton and CyanoHABs. The Final Environmental Impact Statement (FEIS) for the Diamond Lake Restoration Project advocated reducing Tui Chub fish biomass in Diamond Lake (natural lake) as the preferred restoration alternative (Available online: http://www.dfw.state.or.us/fish/local_fisheries/diamond_lake/restoration-update.asp; accessed: 28 May 2019). However, dam releases from Diamond Lake, beginning in 2006 resulted in nutrients impacting the summer trophic status of the downstream portion of the North Umpqua River, and Lemolo Lake (impoundment) (Figure 5).

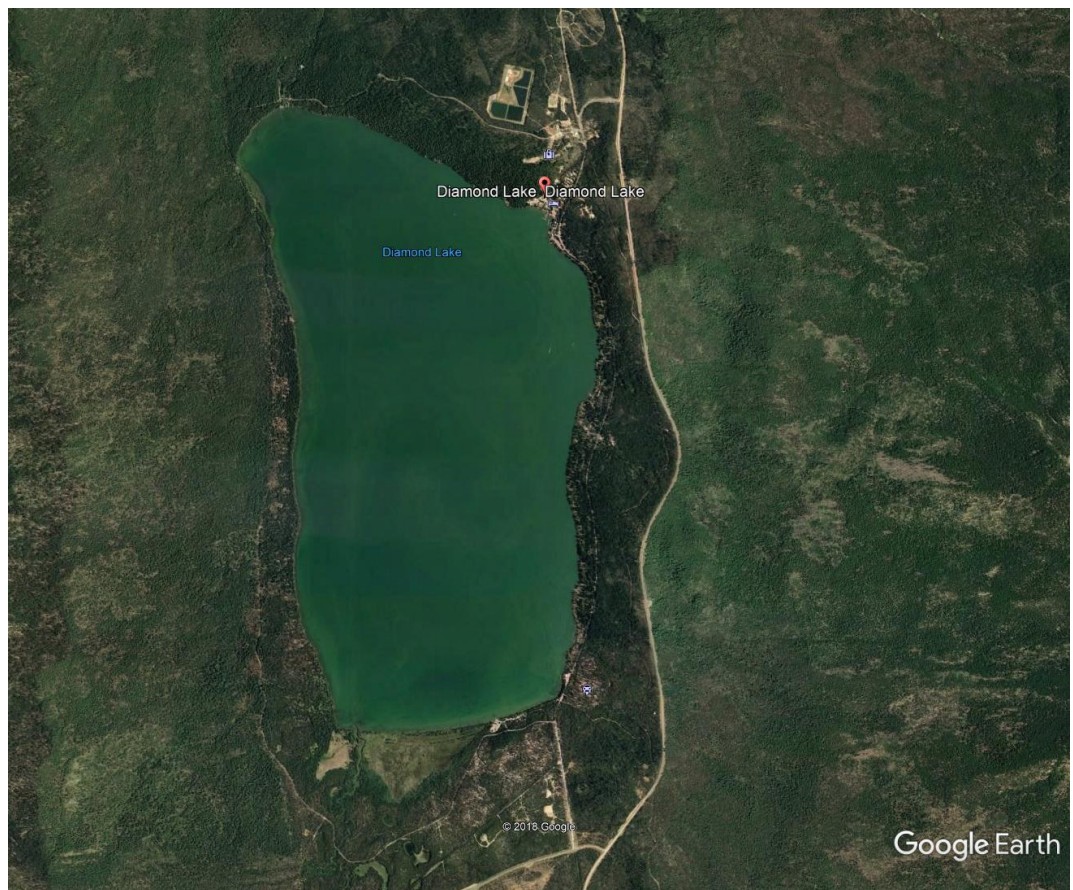

**Figure 4.** Satellite image of Diamond Lake, Oregon. Image taken 6 July 2014. Source: Image obtained from Google Earth.

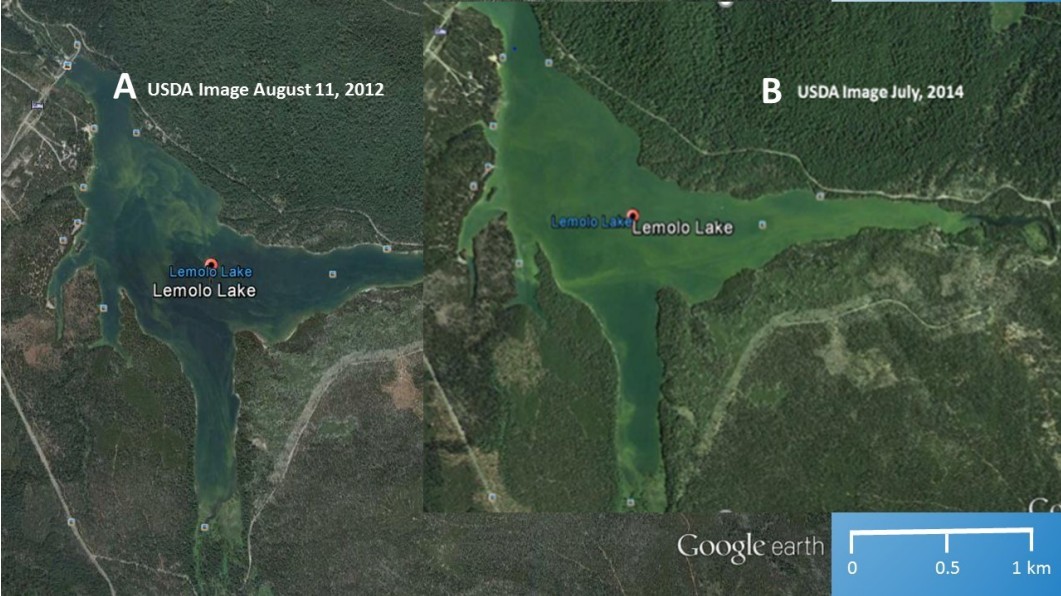

**Figure 5.** USDA color aerial images of Lemolo Lake, Oregon. Images were taken: (**A**) 11 August 2012 and, (**B**) July 2014. Bright green areas within Lemolo Lake (image B) indicates an extensive cyanobacterial bloom occurring at the time the image was taken. Source: Images obtained from Google Earth.

Lemolo Lake, which is the upper-most impoundment on the North Umpqua River, is downstream from Diamond Lake (Figure 5). Hydrodynamic modeling [46,48] of the reservoir indicated an unusual

mixing pattern, created by two stream inlets, (Diamond Lake and North Umpqua River), with greatly differing summer temperatures and nutrient loads (i.e., warmer waters from Diamond Lake, and colder phosphorus-enriched water from the North Umpqua River [48]). As seen in Figure 6, the number of days Lemolo Lake was under a CyanoHAB health advisory increased from 2006 to 2008, during and after the implementation of the USFS Diamond Lake restoration plan to eradicate the Tui Chub in 2006 [47].

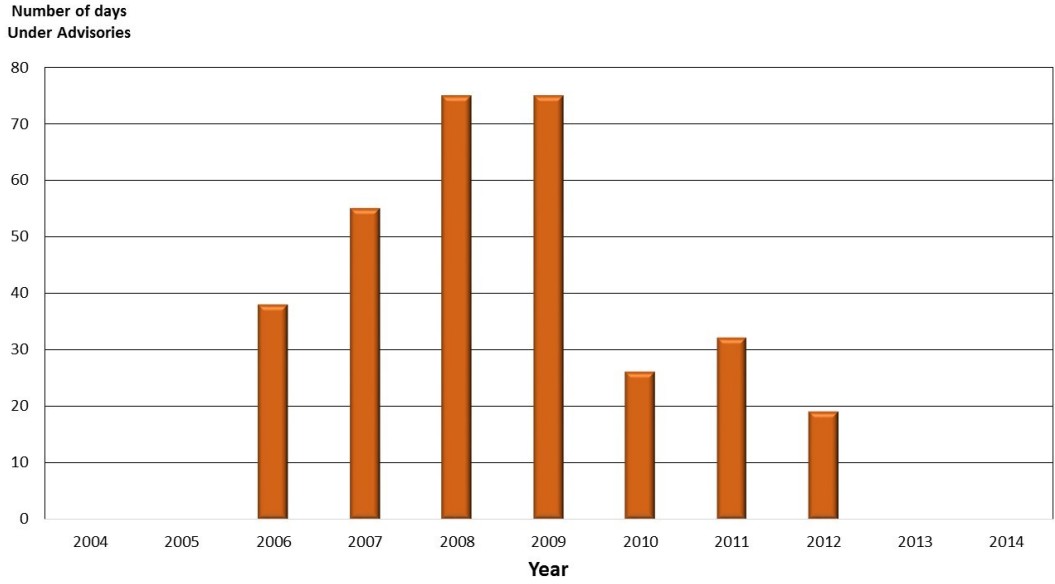

**Figure 6.** Oregon Department of Environmental Quality's (ODEQ) harmful cyanobacterial bloom (CyanoHAB) advisories for Lemolo Lake, and the number of days under an advisory. Current (post-2014) data has not been posted on the website: (Available online: http://www.oregon.gov/oha/ph/HealthyEnvironments/Recreation/HarmfulAlgaeBlooms/Pages/index.aspx; accessed on 28 May 2019). Note: There were no CyanoHAB advisory days for Lemolo Lake in 2004, 2005, 2013, and 2014.

2.1.2. The Tenmile Lakes

The cause of the CyanoHABs in the Tenmile Lakes is similar to Lemolo Lake. The dominant species of cyanobacteria in the Tenmile Lakes are *Microcystis aeruginosa*, *Aphanizomenon flos-aquae*, and *Dolichospermum planctonicum (syn: Anabaena planctonica)* [29]. The Tenmile Lakes, consisting of North Tenmile and Tenmile Lake, are located on the south-central Oregon Coast (Figure 2). The lakes are highly productive coastal fisheries. However, salmonid populations have been declining [40]. Land use in the Tenmile Lakes watershed is predominantly urban along the shoreline of the lakes, agriculture (hay and grazing) in the flood plains of low gradient stream reaches, and logging in the upper watershed. The urban water management approach for the Tenmile Lakes is outlined in [49], which is accomplished through developing improvements to existing urban runoff control structures, and through the restriction of higher-density urban development. The Tenmile Lakes serve as the primary drinking water supply for lakeshore residents. Since the mid-1980s, the drinking water supply has been infused with high populations of cyanobacteria, *Microcystis aeruginosa*, and listed on the Oregon 303(d) list of impaired surface waters [40]. In 1997, Tenmile Lake was temporarily closed as a source of potable water. From 1997 to 2006, the cyanobacteria and toxin levels triggered five health advisories (Figure 7) related to lake water consumption (drinking water) and/or recreational contact with lake waters [40]. Oregon has continued to maintain a CyanoHAB Surveillance Program, and issues CyanoHAB health advisory guidelines on a regular basis [36].

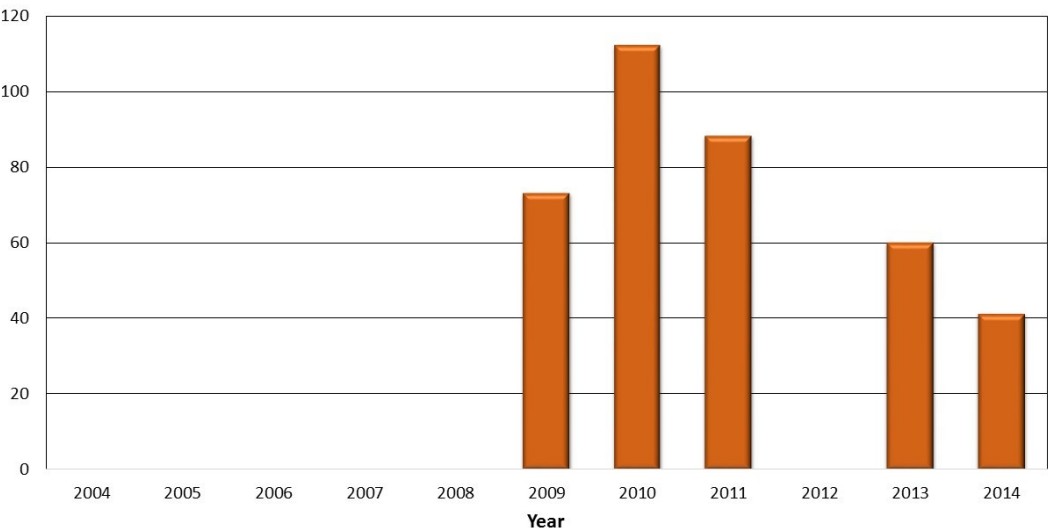

**Figure 7.** Oregon Department of Environmental Quality's (ODEQ) harmful cyanobacterial bloom (CyanoHAB) advisories for Tenmile Lakes and the number of days under an advisory. Note: There were no CyanoHAB advisory days for Tenmile Lakes in 2004, 2005, 2006, 2007, 2008, and 2012.

ODEQ [40] points out that water quality factors affecting weed and cyanobacterial growth occur in the presence of excessive nutrients and sediment. ODEQ [40] describes sediment lake core samples as having increased sediment accrual rates (above natural conditions). The continued water quality and fisheries problems in the Tenmile Lakes have prompted resource management agencies and the City of Lakeside to address the issues.

ODEQ maintains a list of total maximum daily loads (TMDLs) in Oregon approved by The United States Environmental Protection Agency ((EPA)—Available online: https://www.oregon.gov/deq/wq/tmdls/Pages/TMDLs-Approved-by-EPA.aspx; accessed: 28 May 2019). Sediment load is also linked to CyanoHABs, as phosphorus has an affinity for binding with fine particulate matter. Therefore, reducing nitrogen and phosphorus (i.e., sediment) delivery to the Tenmile Lakes is essential. The Tenmile Lakes sediment core samples [42] indicate increased sediment accrual rates began shortly after homesteading began in the area in the mid-to-late 1800s. ODEQ [42] describes sediment accumulation rate increases, with logging and clearing of farm and ranch land, in the 1800s. Logging, railroad, and road construction, occurred from 1910–1920, prior to riparian and water protection measures [42]. From the 1920s into the early 1940s, there was continued wetland-to-agriculture conversion, and timber harvesting. Post-World War II, 1945–1955, saw a surplus of wood products, resulting in a slowdown of timber harvesting. During this period, the ecosystem had a chance to recover. From 1955 to 1999, favorable economics increased timber harvesting activity, and accelerated urban and residential lakeshore development around the Tenmile Lakes, with the drainage of wetlands [42]. A plausible outcome of this increased engineered activity would be increased runoff and sediment load from the logged areas and roads. Adding to this scenario, as seen in Figures 8 and 9, Roberts Creek was moved to the south side of the valley from its original position in the center (Note: the dark spectral signature in center of the valley). Stream channel instability, and incision, could have occurred from altering the channel on the downslope side (south) of the flood plain to capture flood irrigation tail-water. The internal (sediment) loading of nutrients in the three lakes in the study area is greater than the nutrient loading received from watershed inputs [48]. Meeting the required internal loading levels led to the decision to eliminate 90–100% of the Tui Chub population, along with increased monitoring of the trout population to determine if a reduction in the trout population was warranted. The internal loading associated with the Tui Chub was estimated as being four times that of external loading.

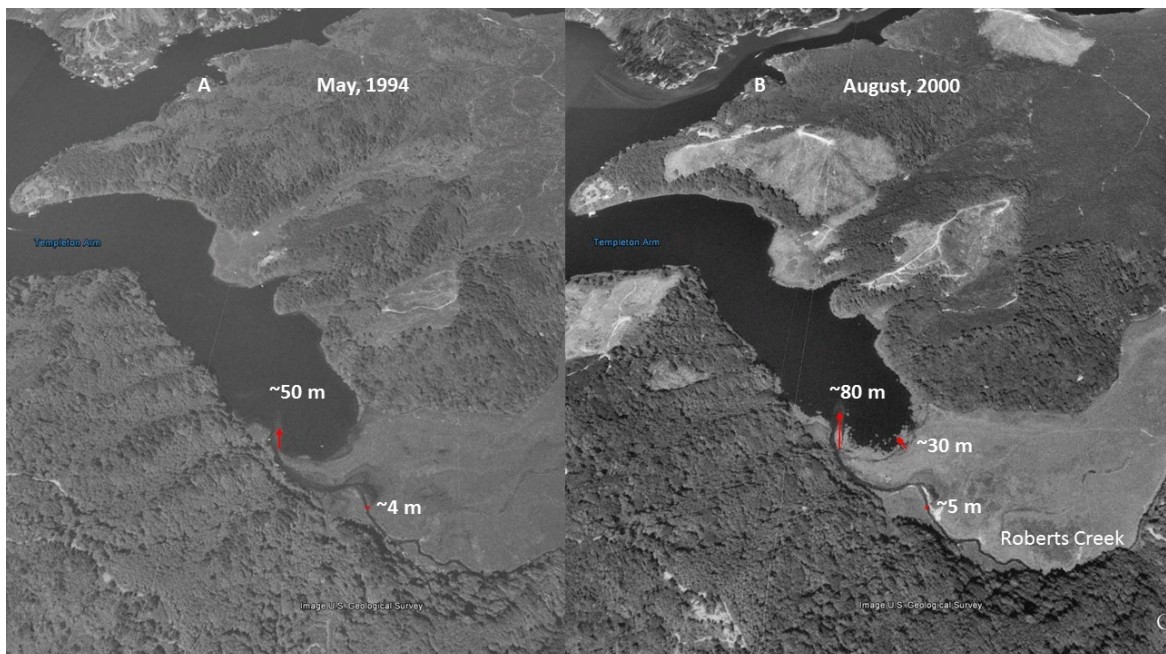

**Figure 8.** US Geological Survey aerial images of Tenmile Lake, Templeton Arm, and Roberts Creek. (**A**): The May 1994, black and white aerial image shows the Roberts Creek channel is incised, and the delta is expanding. (**B**): The August 2000, black and white aerial image shows the Roberts Creek channel has expanded (Note: the white area indicates stream bank slumping), and the delta has extended into the lake approximately 30 m farther than the May 1994 image. It also shows increased clear cuts on the slopes in three areas north of (above) the lake. Source: Images obtained from Google Earth. Note: Red arrows show the extent of delta growth.

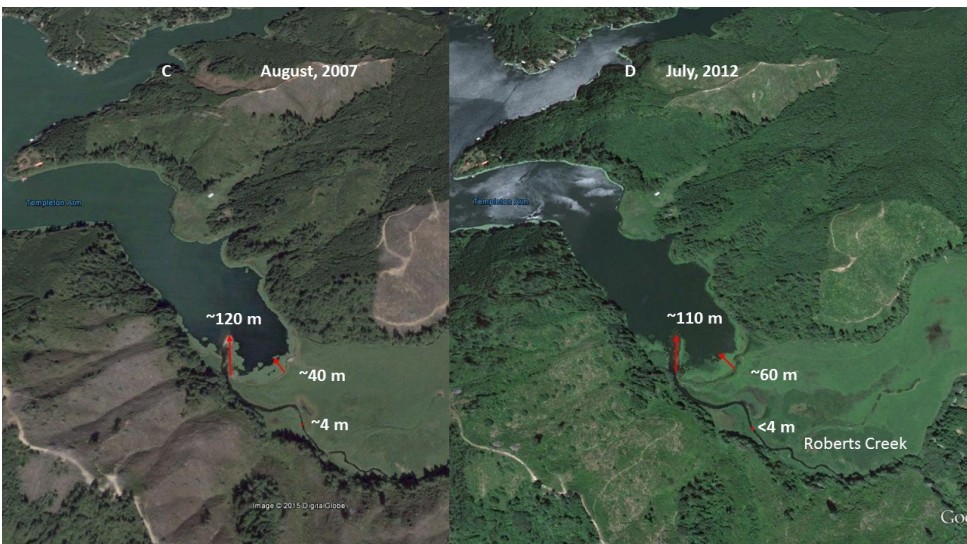

**Figure 9.** US Geological Survey color aerial images of Tenmile Lake, Templeton Arm, and Roberts Creek. (**C**): The August 2007, aerial image shows the Roberts Creek channel is incised, and the delta is expanding. (**D**): The July 2012, aerial image shows the Roberts Creek channel is unchanged, and the delta has been reduced by approximately 10 m from the August 2007 image. The August 2007 image also shows increased clear cuts on the slopes in three areas south of (below) the lake. Source: Images obtained from Google Earth. Note: Red arrows show the extent of delta growth.

Flood irrigation of the meadow required building a diversion dam, and a channel, to flood the upslope side of the field (Figures 8 and 9—north side), consequently incising Roberts Creek, for the

irrigation tail-water to run into (Figures 8 and 9—south side). Lowering the base of the channel caused incision, with a nick point or nick zone, to migrate up the river and tributaries, thus disconnecting the stream from its flood plain. Channelization, and the incising of Roberts Creek, funnels the flow within the channel, resulting in an increased peak flow, causing further incision and bank erosion. Incision, with decreased floodplain access, greatly diminishes sediment and nutrient capture during floods, and greatly diminishes nutrient uptake by riparian vegetation, and the denitrification process [10].

The increased sediment load and dominance of invasive fish species (e.g., Tui Chub, bass) in the Tenmile Lakes have vastly altered the historic condition of the coastal lake ecosystem [42]. These factors indicate changes in the functional condition (i.e., physical processes) of the lakes and watershed, resulting in decreased water quality, loss of salmonid habitat, altered aquatic food chain dynamics, and associated increased CyanoHABs (Figures 8 and 9).

The nutrient report [50], and the TMDL study [37], discusses the importance of restoring and improving stream and wetland riparian functions. Nevertheless, there is no mention of improvement in ecosystem functions. To improve stream and wetland riparian functions, the focus should be on restoring the hydrological connection of the wetlands, to filter and store sediment, prior to entering the lake. The proper ecological, hydrological, and geomorphological functioning of lake and river catchments is important, and including this in the total catchment management activities to protect water quality and reduce CyanoHABs is essential.

## 3. Results: Ecosystem Function and Best Management Practices (BMPs)

Maintaining viable, economically sustainable, forest or agricultural land, is a key element in any federal, state, or local water quality protection program. Implementing a forest, or grazing management plan, and measures to minimize water quality impairment due to forest harvesting and associated activities, is the responsibility of the owners of the land [51]. Therefore, land management measures to reduce upland (i.e., forest, agricultural) NPS runoff should focus on restoration of ecosystem physical functions [9]. As noted in Reference [12], determining the physical processes driving upland and stream and wetland riparian ecological functions enables stakeholders to identify sources and severity of impairment to predict, and avoid, a decline in water quality.

Ecosystem function is notably distinct from other water quality management tools [51]. As seen in Figure 9D, it is difficult to determine which best management practices (BMPs) were applied to the logged area (e.g., replanting, road removal). Because of this, other land managers (private, public) require extra steps to determine if the BMPs used could be replicated on their site. To improve BMP sharing, the USEPA maintains a database that tracks restoration projects using the Clean Water Act Section 319 (CWA 319) NPS project tracking system called the Grants Reporting and Tracking System (GRTS—Available online: https://iaspub.epa.gov/apex/grts/f?p=grts:95; accessed: 28 May 2019). GRTS is the central oversight, management, and data-sharing tool for the Clean Water Act's Section 319 Program. All data in GRTS is entered by state, territory, or tribal agencies receiving Section 319 grants. Unfortunately, for the Tenmile Lakes, the GRTS CWA 319 tracking system did not have information on the BMPs used.

After implementation of the Tui Chub eradication program, the number of days Lemolo Lake was under an advisory were increased in 2007, peaking in 2008 and 2009. In 2010, the number of days under a health advisory was reduced by almost two-thirds (Figure 6). When Diamond Lake was drained, to lower the volume of the lake, and apply less Rotenone pesticide, to eradicate the Tui Chub [45], it had a downstream impact on Lemolo Lake [46,47]. Increased nutrient load into Lemolo Lake overwhelmed the aquatic ecosystem, causing eutrophication, and creating a large area of CyanoHAB (Figure 5), originating from the phosphorus-rich Upper Umpqua River.

In Diamond Lake, during the second year of the Tui Chub eradication program (2007), dramatic improvements were noted in water clarity and aquatic community populations [52]. In 2009, the cyanobacterial level had further reduced to a point that Diamond Lake could meet its water quality standards [52]. As seen in Figure 5, Lemolo Lake would take a few more years to meet that goal.

Ecosystem resiliency is the ability of the local ecology to respond in a positive manner to an acute and/or chronic stressor. Maintaining healthy aquatic and riparian habitats depends on adaptive "management" strategies, allowing for, or facilitating, natural recovery of lentic and lotic functions. In the case of Diamond Lake, the Oregon Department of Fish and Wildlife (ODFW) and the USFS implemented a management plan to eradicate the Tui Chub and save the lake. For the rest of the North Umpqua watershed, changes in the USFS management plan, from 1955 to the present, had the impact of restoring upland, and riparian ecological functions, in the other parts of the watershed. By 1998 [45], lotic ecosystems were close to, or in, proper functioning condition (PFC). As seen in Figure 6, the net result is that the number of days under a CyanoHAB health advisory were reduced to zero by 2013. The added impact was the decrease in sediment, and phosphorus, into the North Umpqua River and Lemolo Lake, as the logging scars healed [44]. As seen in Figure 3B, Lemolo Lake was eutrophic, and had CyanoHABs in the late summer, but CyanoHABs were not reaching a level where there was harm to humans, and therefore requiring CyanoHAB health advisories.

As displayed in a May 1994 black and white (BW) aerial image (Figure 8A), the physical condition of Roberts Creek had an incised stream channel, with an increasing delta, into the Templeton Arm of Tenmile Lake. Delta building indicates two potential sources of sediment: stream channel incision and subsequent channel evolution [14], and other potential land management impacts (e.g., grazing, timber harvesting, and road construction in the harvested areas). In an August 2000 image (Figure 8B), the delta extended approximately 30 m further than in May 1994. The stream channel had widened, and matted CyanoHABs were present, and extending out from the shore. The increase in the delta from 1994 to 2000 matched the sediment increase in the Tenmile Lake core during the same period [37].

As shown in the August 2007 USGS color aerial images of Roberts Creek (Figure 9C), the delta extended approximately 70 m further into the lake than in 1994. In Figure 8B, logging occurred along the north side, and west end, of the Templeton Arm of Tenmile Lake. In 2007 (Figure 9C), logging extended along the length of the Templeton Arm, and into the lower Roberts Creek catchment. The spatial area of the CyanoHAB was larger, and extended further along the shoreline than in August 2000 (Table 2). The July 2012 color aerial image (Figure 9D) shows that the Roberts Creek channel was unchanged, and the delta had decreased in length, by approximately 10 m, from the August 2007 image (Table 2). However, CyanoHABs were much more intensive, and occupy large portions of the shoreline. By 2012, the logged areas were revegetating, and showing some recovery of upland vegetation.

**Table 2.** Measured distance (m) of United States Geological Survey (USGS) aerial images, using Google Earth, for the Roberts Creek delta, approximate stream channel width, and extent of cyanobacterial growth into Tenmile Lake.

| Feature | May 1994 | August 2000 | August 2007 | July 2012 |
|---|---|---|---|---|
| Delta | 50 m | 80 m | 120 m | 110 m |
| Stream Channel Width | 4 m | 5 m | 4 m | <4 m |
| Cyanobacterial Extent | N/A | 30 m | 41 m | 57 m |

The aerial image, time-series analysis of the upper Roberts Creek shows a reach undergoing a management change (Figure 10). As seen in Figure 10, the potential of this narrow meandering stream is to have hydric soils, stabilizing herbaceous riparian vegetation (sedges, rushes), and to be connected to a broad floodplain. The presence of woody plants indicates that the soils are more oxygenated for this setting, which points to a degraded riparian ecosystem. In 2005, the stream channel was incised, wide, and with woody vegetation (Figure 10A). The floodplain, now a terrace, experienced incursion of upland woody plants and/or willows (Figure 10A). This indicates that the floodplain has been drying out, allowing for the migration of upland plants into the former floodplain. As seen in Figure 10B, the rancher implemented a management plan (building fences) to protect the stream. It appears the rancher removed the woody plants to promote the appropriate herbaceous plant community for hay and forage production.

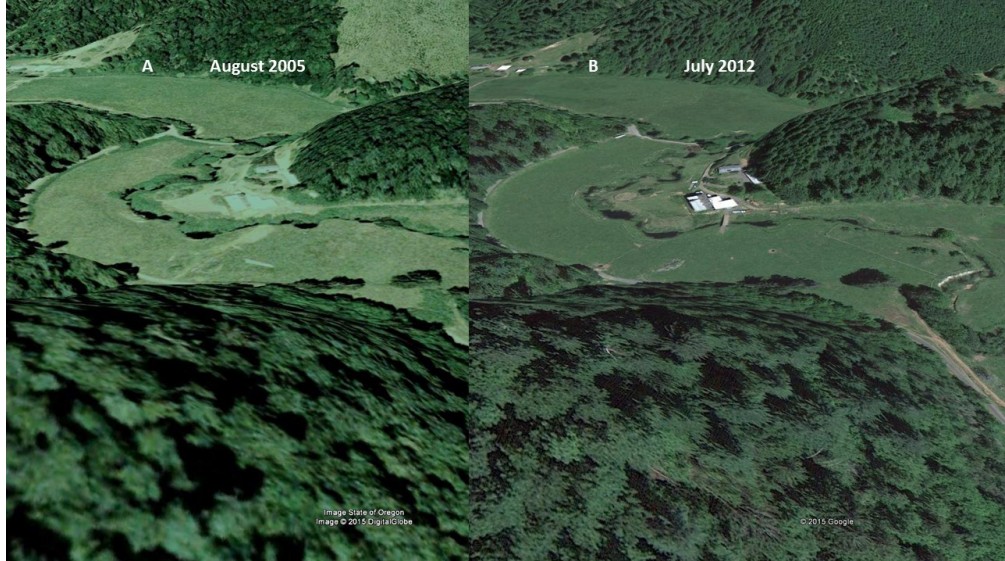

**Figure 10.** (**A**): August 2005 color image of the middle Roberts Creek watershed. Image shows that channel incision has migrated up into the sub-watersheds. The stream riparian area shown on the ranchland was prior to implementation of the management plan. The sub-watershed is dominated by woody plants. (**B**): USDA July 2012 color image after implementation of a natural resources management plan (fencing) for the riparian area. Some woody material has been removed. Source: State of Oregon, DigitalGlobe (2015).

## 4. Discussion

In aquatic environments, not all water pollution is from an external input, which makes attribution of pollution to its source/cause difficult, and managing aquatic ecosystems, and their pollution, problematic. The state of Oregon provides guidance for: classification and profiling of wetland and riparian locations [53], wetland monitoring and assessment [54], and a wetland assessment protocol [55], which provides considerable assistance to those tasked with managing aquatic ecosystems in the state of Oregon. Lewis and Wurtsbaugh [33] indicate reducing phosphorus is essential in curtailing the formation of CyanoHABs. Because of the unique relationship between phosphorus and sediment [56], the best method for reducing phosphorus loads is to reduce sediment from entering the lentic or lotic ecosystem. Sediment is a major pollutant across the United States [7], and frequently surges, in conjunction with functional ecosystem decline. The internal (sediment) loading of nutrients in the three lakes in the study area is greater than the nutrient loading received from watershed inputs [44], and was mitigated by the reduction of the Tui Chub population.

Vegetation provides roughness that slows water velocity. Dissipated energy is less likely, at any one spot, to exceed the critical shear stress of stream bank soils and stream substrate and cause erosion [57]. Similarly, when water slows, it deepens and spreads across accessible floodplain areas, where it slows further from friction with floodplain vegetation, and no longer has the velocity and turbulence to keep particles suspended. Reduced erosion and sediment deposition have a direct impact on water quality [10]. With the loss of ecological functions, stream and wetland riparian areas, and the aquatic environment, converts from a sink to a source of water pollution [58]. Ecological function potential is based on a concept of dynamic equilibrium in an ecosystem, that corresponds to measures of the physical setting [22,59–64]. Leading indicators are capable of measuring alterations to stream and wetland riparian functions [10], which provide essential ecosystem services [9].

A management response to CyanoHABs has three levels: Immediate—breaking the pathway of exposure by providing clean drinking water—this cannot be sustained indefinitely due to limited funding and uncontrollable water demand; Short-Term—breaking the pathway of exposure by initiating a warning system—this publicizes information on (lagging) indicators (CyanoHABs caused by excess

nutrients) after they are detected; Long-Term—managing ecological functions to reduce transport of excess nutrients, thus preventing potential CyanoHABs. The immediate and short-term responses are unsustainable. However, keeping the flood channel narrow inhibits the processes of "self-healing" needed to restore sediment deposition and nutrient sequestration, and the assimilation processes that are important in improving water quality [10].

The goal of a successful resource management plan is to assist an ecosystem to respond in ways that enhance natural remediation [26,51,65,66], or quicken its pace to a particular level of functionality, or desired condition [9,12,15]. The first step is to assess the potential functioning condition, then determine the primary sources of nutrients. If NPSs are predominant, then it is essential to improve watershed, stream, and wetland riparian functions as part of any watershed management plan for long-term water quality improvement [11,12]. Properly functioning streams and wetland riparian ecosystems provide a steadying influence on water quality and aquatic habitat attributes. Restoring riparian functions will result in slowing the nutrient spiral, with flooding and floodplain deposition, and allow nutrient uptake, aquifer recharge, and reconstruction of quality habitat, and complex niches/food webs that interrelate riparian and aquatic ecosystems [10,67].

This research project illustrates that success in pollution control and in maintaining healthy aquatic habitats often depends on managing land to facilitate natural recovery of riparian functions [15]. Using best management practices throughout the watershed by using a combination of tools and resources, such as: time-series aerial photos (to display changes in overall ecosystem state, and in specific details, due to land use changes), TMDLs (to monitor the amount of nutrients, sediment, and CyanoHABs), CyanoHAB advisories (to inform the public and reduce their exposure to toxins), PFC (to characterize how well physical processes in a stream and wetland riparian area can sustain resiliency), and by addressing upland sources of accelerated nutrient supply to the stream, through appropriate forestry management approaches and road network design, can be applied to positively affect ecological function.

**Author Contributions:** R.K.H., D.T.H., and J.L., designed the project. J.L.A., and R.J.S., performed data analysis. E.S.H., and R.K.H., wrote and edited the manuscript. S.S., M.J.P., T.J.-L., E.W., and H.K., reviewed the manuscript. E.S.H., had the manuscript internally peer-reviewed, performed the final edits, and approved the final manuscript.

**Funding:** This research was funded by the United States Environmental Protection Agency, through its Office of Research and Development, under contract EP-15-Z-000082 to Dr. Joan L. Aron, and contract EP-14-Z-000031 to Dr. Robin J. Schafer.

**Acknowledgments:** The authors would like to thank Patti Tyler, USEPA Region 8, David Guiliano, USEPA Region 9, Adam Jorge, USEPA, Yongping Yuan, USEPA, Nilla Barros, USEPA, and Joyce Swanson for their critical review of this manuscript. Although this paper has been subjected to USEPA review and is approved for publication, it may not necessarily reflect official Agency policy. Mention of trade names and commercial products does not constitute endorsement or recommendation for use.

**Conflicts of Interest:** The authors declare no conflict of interest.

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
