# Peer review of "An Ecological Function Approach to Managing Harmful Cyanobacteria in Three Oregon Lakes: Beyond Water Quality Advisories and Total Maximum Daily Loads (TMDLs)"

_water, doi:10.3390/w11061125_

Round 1
Reviewer 1 Report
The manuscript addresses an important and topical issue of water quality monitoring in lakes (and running waters) as well as water quality “management” at the Oregon Department of Environmental Quality (ODEQ). The Abstract and a comprehensive reference list suggest that the manuscript will be very interesting – in particular for professionals dealing with analyses and assessments of the ecological status of water bodies. Unfortunately, the Authors do not provide any information on the criteria used to evaluate water quality or quality management methods. Moreover, the description of research methods (e.g. Proper Functioning Condition (PFC) stream and wetland riparian area assessment protocol) does not correspond with the results of the study. The Results section presents satellite images of forest thinning and clear cutting areas in the catchment of Lake Lemolo, and the difference in phytoplankton blooms between August 2012 and July 2014 in Lake Lemolo, diagrams showing “the number of days under an advisory” in Lakes Lemolo and Tenmile, as well as changes in “delta” length, channel stream width and the extent of algal growth expressed in meters (?) in Lake Tenmile. The second part of the Results section, entitled “Results: Ecosystem Function and Best Management Practices (BMPs)”, placed after the Discussion section (!), presents two satellite images illustrating changes “after implementation of a natural resources management plan (fencing) for the riparian area” in the middle Roberts Creek watershed. The manuscript ends with a “Conclusion” section which is 9 lines (more than 20%) longer than the Discussion section. In my opinion, the paper makes a confused and fragmented impression; it is not well structured. The manuscript is too lengthy (18 pages) and does not contain valuable information about the methods for assessing and managing the quality of aquatic ecosystems in the State of Oregon, which could be very interesting to readers. The above shortcomings cannot be compensated by the correct assumptions for the ecosystem monitoring and management project, described in rather general terms.
Author Response
Unfortunately, the Authors do not provide any information on the criteria used to evaluate water quality or quality management methods.
Agree: The manuscript was revised in the following ways: a references [76], and [77], were added in Section 2.1, Second Paragraph, Fourth Sentence, to note water quality management plans that were used in the study area, and the following text was added to the end of that sentence, “… , and this points to the importance of having a water quality management plan, as was developed for the Tenmile Lakes watershed [76], Diamond Lake and Lemolo Lake [77], to mitigate those pollutant sources.”
Moreover, the description of research methods (e.g. Proper Functioning Condition (PFC) stream and wetland riparian area assessment protocol) does not correspond with the results of the study.
Agree: The manuscript (Lines 188 – Line 193) was revised, with additional text, to read as follows, “The Oregon DEQ has well-developed surveillance and monitoring program, in coordination with educational outreach, waste water permitting, water quality and drinking water quality standards, and health advisories for HABs, linked with TMDL approach. This was augmented with a Proper Functioning Condition (PFC) stream and wetland riparian area assessment protocol [16], [21], conforming to the United States Department of Agriculture’s (USDA’s) United States Forest Service (USFS) process.”
The Results section presents satellite images of forest thinning and clear-cutting areas in the catchment of Lake Lemolo, and the difference in phytoplankton blooms between August 2012 and July 2014 in Lake Lemolo, and diagrams showing “the number of days under an advisory” in Lakes Lemolo and Tenmile, as well as changes in “delta” length, channel stream width and the extent of algal growth expressed in meters (?) in Lake Tenmile.
Figure 5 has a legend to provide the extent of the algal growth, delta length, and channel stream width expressed using 1 kilometer as the base measurement, due to the size of these features.
The second part of the Results section, entitled “Results: Ecosystem Function and Best Management Practices (BMPs)”, placed after the Discussion section (!), presents two satellite images illustrating changes “after implementation of a natural resources management plan (fencing) for the riparian area” in the middle Roberts Creek watershed.
Results section has been moved ahead of Discussion section.
The manuscript ends with a “Conclusion” section which is 9 lines (more than 20%) longer than the Discussion section. In my opinion, the paper makes a confused and fragmented impression; it is not well structured. The manuscript is too lengthy (18 pages) and does not contain valuable information about the methods for assessing and managing the quality of aquatic ecosystems in the State of Oregon, which could be very interesting to readers. The above shortcomings cannot be compensated by the correct assumptions for the ecosystem monitoring and management project, described in rather general terms.
The Conclusion has been shortened to 387 words.
Agree: The manuscript was revised in the following way, the first sentence in the Discussion section now has information on the guidance that Oregon provides for managing aquatic ecosystems, and now reads as follows, “In aquatic environments, not all water pollution is from an external input, which makes attribution of pollution to its source/cause difficult, and managing aquatic ecosystems, and their pollution, problematic. The state of Oregon provides guidance for: classification and profiling of wetland and riparian locations [72]; wetland monitoring and assessment [73], and; a wetland assessment protocol [74], which provides considerable assistance to those tasked with managing aquatic ecosystems in the state of Oregon.”
The manuscript length has been reduced by shortening Section 3 (Results), Section 4 (Discussion), and Section 5 (Conclusion), along with the Abstract, Introduction, and Methods Section. The entire manuscript is now 4 pages shorter.
Reviewer 2 Report
I found this manuscript difficult going and hard to read. It read more like a government departmental report rather than a paper intended for publication in an international scientific journal. I found parts of it long-winded and some parts repetitious. I thought that the organisation was poor, with some results presented in Section 2 (Methods), and more results in Section 4 (Results: Ecosystem Function and Best Management Practices), which came after the Discussion. Also I thought that it was a bit too US orientated - parts may be of interest to US readers, but of less relevance to an international audience (e.g. lines 470-479, but also elsewhere).
That said, I think that, although very descriptive, from the results shown, especially the time series aerial photographs, it tells a rather nice story that is well worth telling. Also, I think that the proper ecological, hydrological and geomorphological functioning of lake and river catchments is important and including this in total catchment management activities to protect water quality and reduce harmful cyanobacterial blooms further downstream is well worth emphasising to other readers.
In short, I think that the paper is worth publishing, as it provides a good case study of how land use impacts affect lakes and streams. It also shows what active catchment management can do to produce positive water quality results. However it needs to be considerably shortened, better structured and less repetitious. At present, it is a good story spoiled by poor story telling.
I found that the Abstract was too long, verbose and with too much on TMDLs and their inadequacies, This (and the title) led me to initially believe that the paper was mainly on TMDLs, which it was not. The abstract needs to focus more on what is in the paper itself. Perhaps discussion of TMDLs could be in the "Discussion" section of the paper (there is some brief discussion, and a paragraph at the end of "Conclusions"), or not mentioned at all?
Section 1.2. Maybe just concentrate on cyanobacterial toxins in freshwaters as the paper is about freshwater lakes and their catchments (not about marine algal toxins). Line 133 - "cyanotoxins can inhibit zooplankton grazing" ?? Line 141 Microcystis spp. (Microcystis are cyanobacteria, not algae).
Line 149. Is this referring to cyanobacterial biomass strongly correlated with TP? Line 153, 339. Not all cyanobacteria fix atmospheric nitrogen.
Line 185- 187. Confusing, maybe contradictory. "qualitative assessment process", "consistent, quantitative, science-based approach"???
Fig 2. Maybe a map would better show the lakes and the rivers and streams connecting them and in their catchment areas?
Table 1. Minimum depth ?? Should be zero? Maybe is meant to be mean depth?
Lines 260- 267. Which lakes are impoundments, which are natural?
Section 2.1.1. Throughout section there is mention of lakes being "naturally high in phosphorus", "a nitrogen-limited eutrophic lake", "increased nutrient load....causing eutrophication" "dramatic improvements were noted in water clarity", "decrease in sediment and phosphorus" - yet there are no data given. These may perhaps be found in the other literature cited, but it is unclear. Some actual data would assist, not just citations to data in other publications.
Figure 6. Number of advisories are not shown in the figure, but mentioned in the caption.
Lines 314-315. What urban water management is undertaken, to ensure Tenmile Lake is not impacted by urban wastewater or stormwater runoff? Urban impacts may be greater than rural (agricultural, forestry) impacts ??
Page 12. How much internal loading occurs in these lakes, with nutrients coming from sediments already in the bottom of the lakes? Is this an issue? (See also 3. Discussion - some mention of internal loads would be useful here). Does irrigation tailwater flow into the lake? Removal of wetlands would likely increase nutrient and sediment inputs into the lakes. What are the algal mats in Tenmile Lake? Filamentous green algae such as Cladophora or Stigeclonium, etc.? Or filamentous benthic cyanobacteria, such as Oscillatoria or Phormidium?
I think that Sections 3, 4 and 5 describe the outcomes of the study adequately and the importance of using drivers of ecological function as a key management tool, although maybe they could be shortened a little. The final paragraph Lines 530-548) about TMDLs is questionable as only a little about these and how they were applied to the study watershed has been discussed previously in the manuscript.
References. There are a heck of a lot of references (up to 123) included in the reference list, but I could find only up to number 71 in the manuscript text. Are all these necessary? Many also appear to come from the grey literature.
I would recommend that the paper be published after considerable revision, in a shorter more concise form.
Author Response
I found this manuscript difficult going and hard to read. It read more like a government departmental report rather than a paper intended for publication in an international scientific journal. I found parts of it long-winded and some parts repetitious. I thought that the organisation was poor,
Agree: The manuscript has been reorganized, and the writing made clearer to make the text easier to read.
Results section has been moved ahead of the Discussion section.
The manuscript length has been reduced by shortening Section 3 (Results), Section 4 (Discussion), and Section 5 (Conclusion), along with the Abstract, Introduction, and Methods Section. The entire manuscript is now 4 pages shorter.
All of the multiple instances of repetitive text in the manuscript have been removed.
with some results presented in Section 2 (Methods), and more results in Section 4 (Results: Ecosystem Function and Best Management Practices), which came after the Discussion. Also I thought that it was a bit too US orientated - parts may be of interest to US readers, but of less relevance to an international audience (e.g. lines 470-479, but also elsewhere).
The results listed in the Methods section (Lines 278 – 286; Lines 294 – 297, and; Lines 380 – 398) have been moved to the Results section.
Results section has been moved ahead of the Discussion section.
In Section 1.2, last sentence, a reference [35], notes that harmful algal blooms are rising in predominance globally. This is not a change, and was in the manuscript as originally submitted.
Also, I think that the proper ecological, hydrological and geomorphological functioning of lake and river catchments is important and including this in total catchment management activities to protect water quality and reduce harmful cyanobacterial blooms further downstream is well worth emphasising to other readers.
The following was added as the last sentence in Section 2.1.1, “The proper ecological, hydrological and geomorphological functioning of lake and river catchments is important, and including this in the total set of catchment management activities, to protect water quality and reduce harmful cyanobacterial blooms, is essential.”
In short, I think that the paper is worth publishing, as it provides a good case study of how land use impacts affect lakes and streams. It also shows what active catchment management can do to produce positive water quality results.
Thank you very much for the vote of confidence.
However it needs to be considerably shortened, better structured and less repetitious. At present, it is a good story spoiled by poor story telling.
Results section has been moved ahead of the Discussion section.
The manuscript length has been reduced by shortening Section 3 (Results), Section 4 (Discussion), and Section 5 (Conclusion), along with the Abstract, Introduction, and Methods Section. The entire manuscript is now 4 pages shorter.
All of the multiple instances of repetitive text in the manuscript have been removed.
I found that the Abstract was too long, verbose and with too much on TMDLs and their inadequacies, This (and the title) led me to initially believe that the paper was mainly on TMDLs, which it was not. The abstract needs to focus more on what is in the paper itself. Perhaps discussion of TMDLs could be in the "Discussion" section of the paper (there is some brief discussion, and a paragraph at the end of "Conclusions"), or not mentioned at all?
The Abstract has been shortened to 228 words.
Agree: The manuscript was revised in the following way: the following sentence was added in Section 1.3 as the next-to-last sentence, to clarify the fact that TMDLs are important, and should be used within an overall framework using complementary tools, techniques, and approaches, including PFC, to enhance its usability, and not as the sole methodological approach for managing cyanobacteria, “This study recommends using the Oregon DEQ TMDL approach, in conjunction with, time-series photography, HAB advisories, and the PFC methodology, as a preferred (best) practice, instead of using TMDLs without complementary approaches.”
The text discussing TDMLs at the end of the Conclusion section has been moved to the Discussion section just before the last paragraph. The text discussing TDMLs in the abstract, with the exception of the first two mention of TDML in the abstract used, which set the context for the manuscript, has been moved to the Discussion section just before the last paragraph.
Section 1.2. Maybe just concentrate on cyanobacterial toxins in freshwaters as the paper is about freshwater lakes and their catchments (not about marine algal toxins).
All references to ‘marine’ have been removed from the manuscript.
Line 133 - "cyanotoxins can inhibit zooplankton grazing" ?? added reference [75].
Line 141 Microcystis spp. (Microcystis are cyanobacteria, not algae). Changed word algae to cyanobacteria.
Line 149. Is this referring to cyanobacterial biomass strongly correlated with TP? Yes.
Line 153, 339. Not all cyanobacteria fix atmospheric nitrogen.
Changed line 153 to: “…special nitrogen fixing properties of some species of cyanobacteria (e.g., aerobically [Anabaena, Gloeothece, Cyanothece, Lyngbya, Trichodesmium, and Katagnymene]; anaerobically or microaerobically [Plectonemaboryanum] - https://www.intechopen.com/books/advances-in-biology-and-ecology-of-nitrogen-fixation/nitrogen-fixing-cyanobacteria-future-prospect, Accessed: 28 April 2019)…”
Changed line 339 to: “…special nitrogen fixing properties of some species of cyanobacteria (e.g., aerobically [Anabaena, Gloeothece, Cyanothece, Lyngbya, Trichodesmium, and Katagnymene]; anaerobically or microaerobically [Plectonemaboryanum] - https://www.intechopen.com/books/advances-in-biology-and-ecology-of-nitrogen-fixation/nitrogen-fixing-cyanobacteria-future-prospect, Accessed: 28 April 2019)…”
Line 185- 187. Confusing, maybe contradictory. "qualitative assessment process", "consistent, quantitative, science-based approach"??? Changed to: “…PFC assessment framework is a consistent, qualitative, science-based approach for considering…”
Fig 2. Maybe a map would better show the lakes and the rivers and streams connecting them and in their catchment areas?
The available hydrographical and topographical maps for this region contain too much detail to be useful in this context. Also, adding another figure (or series of figures) including these types of maps would increase the overall length of the manuscript without providing additional insight.
Table 1. Minimum depth ?? Should be zero? Maybe is meant to be mean depth? Changed ‘Minimum’ to ‘Average’, as is listed in the source [45].
Lines 260- 267. Which lakes are impoundments, which are natural?
Line 261, after ‘Diamond Lake’, added, “(natural lake)”;
Line 265, after ‘Lemolo Lake’, added, “(impoundment)”;
Lines 183 and 184, ‘Tenmile Lakes’, added, “(natural lake)”; after ‘Lemolo Lake’, added, “(impoundment)”; after ‘Diamond Lake’, added, “(natural lake)”.
Also, the original manuscript text states in the first sentence immediately under Figure 4, that, “Lemolo Lake is downstream from Diamond Lake, which is the upper-most impoundment on the North Umpqua River (Figure 5).
Section 2.1.1. Throughout section there is mention of lakes being "naturally high in phosphorus", "a nitrogen-limited eutrophic lake", "increased nutrient load....causing eutrophication" "dramatic improvements were noted in water clarity", "decrease in sediment and phosphorus" - yet there are no data given. These may perhaps be found in the other literature cited, but it is unclear. Some actual data would assist, not just citations to data in other publications.
Line 260 - Line 261 was changed to add the following highlighted text, “Diamond Lake (Figure 4), a nitrogen-limited eutrophic lake, peaking at 0.5 milligrams per liter total N, and nearly 0.2 milligrams per liter ammonium-N in 2007 [78], is an important source of water and nutrients to the North Umpqua River [49].”
Line 263 - Line 264 was changed to add the following highlighted text, “Diamond Lake is naturally high in phosphorus, as high as 0.060 milligrams per liter [79].”
Figure 6. Number of advisories are not shown in the figure, but mentioned in the caption.
The number of advisories (units: days) are shown on the left-hand side of the graph in Figure 6.
Lines 314-315. What urban water management is undertaken, to ensure Tenmile Lake is not impacted by urban wastewater or stormwater runoff? Urban impacts may be greater than rural (agricultural, forestry) impacts ?? added reference [76]., and added the following sentence in Line 320 and Line 321, “Urban water management for the Tenmile Lakes is outlined in [76].”
Page 12. How much internal loading occurs in these lakes, with nutrients coming from sediments already in the bottom of the lakes? Is this an issue? (See also 3. Discussion - some mention of internal loads would be useful here). Does irrigation tailwater flow into the lake? Removal of wetlands would likely increase nutrient and sediment inputs into the lakes.
Lines 366 to Line 368 (page 12) - added the following sentence: “The internal (sediment) loading of nutrients in the three lakes in the study area is greater than the nutrient loading received from watershed inputs [47].”;
Discussion – Line 431 – Line 432 - added the following sentence: “The internal (sediment) loading of nutrients in the three lakes in the study area is greater than the nutrient loading received from watershed inputs [47].”
What are the algal mats in Tenmile Lake? Filamentous green algae such as Cladophora or Stigeclonium, etc.? Or filamentous benthic cyanobacteria, such as Oscillatoria or Phormidium?
Added the following sentence to Line 316 and Line 317: “The dominant species of cyanobacteria in the Tenmile Lakes are Microcystis aeruginosa, Aphanizomenon flos-aquae, and Anabaena planctonica [47].”
I think that Sections 3, 4 and 5 describe the outcomes of the study adequately and the importance of using drivers of ecological function as a key management tool, although maybe they could be shortened a little.
The manuscript length has been reduced by shortening Section 3 (Results), Section 4 (Discussion), and Section 5 (Conclusion) has been shortened to 387 words. The entire manuscript is now 4 pages shorter.
The final paragraph Lines 530-548 about TMDLs is questionable as only a little about these and how they were applied to the study watershed has been discussed previously in the manuscript.
Section 1.3 addresses how Oregon’s Department of Environmental Quality uses TMDLs to manage harmful algal blooms. For clarity, Line 177 to Line 179 in Section 3 added the following sentence: “This study recommends using the Oregon DEQ TMDL approach, in conjunction with, time-series photography, HAB advisories, and the PFC methodology, as a preferred (best) practice, instead of using TMDLs without using other complementary approaches.”
References. There are a heck of a lot of references (up to 123) included in the reference list, but I could find only up to number 71 in the manuscript text. Are all these necessary? Many also appear to come from the grey literature.
Agree: The number of references was reduced from 123 to the 79 that were actually cited in the manuscript.
I would recommend that the paper be published after considerable revision, in a shorter more concise form.
The paper has been revised and shortened.
Round 2
Reviewer 1 Report
The correction definitely ordered the narrative and made it easier to understand the text.
Author Response
Reviewer#1: Extensive editing of English language and style required Response: The entire manuscript was re-reviewed, and the language modified to improve readability and meaning of the text. The correction definitely ordered the narrative and made it easier to understand the text. Response: Thank you for the comment.
Reviewer 2 Report
I have reviewed the revised manuscript and find it much improved, although I still find it a hard read, but that could just be me !! I think that the authors have tried to satisfy most of my comments on the original manuscript.
Just a few minor comments.
Line 120 - would still like to see the word "blooms" removed.
Section 1.2. Maybe I am just being trivial, but I find the use of "algae" a poor term here. Cyanobacteria are prokaryotes and not to be confused with eukaryotic "algae". Also maybe instead of "algal toxins" could use "cyanobacterial toxins" or "cyanotoxins".
Line 285. I think this should be Dolichospermum planctonicum. All the taxonomic names should be in italics (throughout paper).
Line 290. This only provides a reference number to urban water management. Does urban runoff and/or wastewater enter the lake, and if so, how is it managed? Maybe a brief sentence, rather than just a reference?
Lines 326-328 and again Lines 465-466. What is being done about the internal loading, if it is greater than the external loading? This will slow the lake remediation progress. If external loading can be greatly reduced, the internal load should eventually diminish also as it is exported or sources are buried deep in the sediments.
Still some repetition throughout the paper. For example, is it necessary to define lentic and lotic every time these terms are used? Providing the explanation in brackets the first time the terms are used should suffice.
The effects of the removal of the zooplanktion eating fish fish in Diamond Lake is interesting, and enhances the effects of catchment management for that lake. A good example of biomanipulation.
A few minor changes perhaps, and the paper shoud be ready for publication.
Author Response
Reviewer#2: I have reviewed the revised manuscript and find it much improved, although I still find it a hard read, but that could just be me !! I think that the authors have tried to satisfy most of my comments on the original manuscript. Response: Thank you for the comment. Just a few minor comments. Line 120 - would still like to see the word "blooms" removed. Response: The word ‘blooms’ was removed, and the sentence now reads, “…algal toxins can inhibit …”. Note: The term ‘harmful algal bloom’ is an official term used by both the US Environmental Protection Agency (EPA), and the US National Oceanic and Atmospheric Administration (NOAA). Also note that references and webpages in the manuscript containing the term ‘harmful algal bloom’ cannot be modified. The places in the manuscript where the term ‘harmful algal bloom’ is used, and defined, and is in the name of a reference or website, have not been changed, however, all instances of ‘algal blooms’ have been replaced by ‘algal toxins’, ‘cyanobacterial blooms’ has been replaced by ‘cyanobacterial toxins’, and ‘phytoplankton blooms’ has been replaced by ‘phytoplankton toxins’. All instances of ‘algal’ in the manuscript have been replaced by ‘cyanobacteria’. Section 1.2. Maybe I am just being trivial, but I find the use of "algae" a poor term here. Cyanobacteria are prokaryotes and not to be confused with eukaryotic "algae". Also maybe instead of "algal toxins" could use "cyanobacterial toxins" or "cyanotoxins". Response: The instances of ‘algae’ have been replaced by ‘cyanobacteria’ in Section 1.2. All instances of ‘algal toxins’ in the entire manuscript have been replaced by ‘cyanobacterial toxins’. Line 285. I think this should be Dolichospermum planctonicum. All the taxonomic names should be in italics (throughout paper). Response: The word “planctonica” is now “planctonicum”. All taxonomic names in the manuscript are now in italics. Line 290. This only provides a reference number to urban water management. Does urban runoff and/or wastewater enter the lake, and if so, how is it managed? Maybe a brief sentence, rather than just a reference? Response: The sentence in Line 290 has added the following text, “, which is accomplished through developing improvements to existing urban runoff control structures, and through the restriction of higher-density urban development.” Lines 326-328 and again Lines 465-466. What is being done about the internal loading, if it is greater than the external loading? This will slow the lake remediation progress. If external loading can be greatly reduced, the internal load should eventually diminish also as it is exported, or sources are buried deep in the sediments. Response A: The following sentence was added at the end of Line 328, “Meeting the required internal loading levels led to the decision to eliminate 90 – 100% of the Tui Chub population, along with monitoring the trout population, to determine if a reduction in the trout population was warranted. The internal load associated with the Tui Chub was estimated as being four times that of external loading.” Response B: The following was added at the end of Line 466, “, and was mitigated by the reduction of the Tui Chub population.” Still some repetition throughout the paper. For example, is it necessary to define lentic and lotic every time these terms are used? Providing the explanation in brackets the first time the terms are used should suffice. Response: The terms ‘lentic’ and ‘lotic’ are now defined only at their first use, and subsequent re-definitions are no longer repeated throughout the manuscript. The effects of the removal of the zooplankton eating fish in Diamond Lake is interesting, and enhances the effects of catchment management for that lake. A good example of biomanipulation. Response: Thank you for the comment. A few minor changes perhaps, and the paper should be ready for publication. Response: All requested changes have been addressed.
